.

# Truly Conserving with Conservative Remapping Methods

Karl E. Taylor

Program for Climate Model Diagnosis and Intercomparison, Lawrence Livermore National Laboratory, Livermore, CA 94550, USA

**Correspondence:** Karl E. Taylor (taylor13@llnl.gov)

**Abstract.**

Conservative mapping of data from one horizontal grid to another should preserve certain integral or mean properties of the original data. This may be essential in some model applications, including ensuring realistic exchange of energy and mass between coupled model components. It can also be essential for certain types of analysis, such as evaluating how far a system is from an equilibrium state. For some common grids, existing remapping algorithms may fail to perfectly represent the shapes and sizes of grid cells, which leads to errors in the remapped fields. A procedure is presented here that enables users to rely on the mapping weights generated by remapping algorithms but corrects for their deficiencies. With this procedure, for a given pair of source and destination grids, a single set of remapping weights can be applied to remap any variable, including those with grid cells that are partially or fully masked.

## 1 Introduction

When analyzing climate data from different sources, it is often necessary, as an initial step, to map the data to a common grid, a procedure commonly referred to as remapping or "regridding" the data. For some purposes it is essential when remapping the data that the global mean (or, alternatively, the global integral) of the field be preserved. Conservative remapping algorithms are meant to guarantee this. In practice, remapping occurs in two steps: 1) given a source and destination grid, mapping "weights" are computed, and then 2) a sparse matrix multiplication of the source data by the weights yields the values of the field on the destination grid. Our focus here is on the second step: given the weights needed for remapping conservatively, we provide guidance on how they should be applied. Appendix A lists some remapping packages that can be used to generate weights (i.e., to execute step 1). It should be noted that nearly all of these packages slightly misrepresent the true shape of grid cells found in some subset of commonly encountered grids. This can cause errors which must be corrected if conservative remapping is demanded. Moreover, most packages provide inadequate guidance on how to handle fields that are partially masked or for three dimensional fields how to account for variations in the thickness of individual layers. The main purpose here is to clearly explain how to compensate for any inaccuracies in a remapping algorithm's representation of grid cell shapes and to account for missing or partially masked data when that is necessary.

The objective in remapping conservatively is to preserve certain physically important characteristics of the climate system. For a climate in global thermodynamic equilibrium, for example, the mean net flux of energy at the top of the atmosphere is

zero. In properly formulated models run with all externally-imposed conditions unchanging over time, the net flux at the top of the atmosphere will indeed approach and fluctuate about zero as the system approaches equilibrium. When the fluxes from a simulation of this kind are mapped to a different grid, we would like to preserve this important characteristic of the simulation. This can only be done if the remapping algorithm is conservative.

As a second example, consider trace species concentration in the atmosphere (e.g., of water vapor, ozone, $CO_2$) or in the ocean (e.g., of salinity, nitrogen, carbon, etc.). When mapping a concentration field to a different grid, it can for some analyses be essential that the total mass of the species of interest be preserved. Similarly, the remapping of sources and sinks of species must preserve the global totals.

The fundamental relationship that must be satisfied to preserve the integral of a field over the global area is:

$$r_{\mathrm{s}}^2 \sum_i F_{\mathrm{s}i} f_{\mathrm{s}i} A_{\mathrm{s}i} = r_{\mathrm{d}}^2 \sum_j F_{\mathrm{d}j} f_{\mathrm{d}j} A_{\mathrm{d}j} \,, \tag{1}$$

where $F$ is the quantity that is mapped from a source grid to a destination grid, $A$ is the grid cell area (expressed as as a solid angle and globally spanning $4\pi$ steradians), $f$ is the grid cell fraction where $F$ is defined (i.e., the "unmasked fraction" of a cell), and the subscripts, s and d, distinguish between the source and destination grids, respectively. The $i$ and $j$ indices apply to source and destination cells, respectively, and the sums are over the entire domain. In certain modeling applications, the radius of the earth, $r$, for the source and destination grids may differ slightly, and this is accounted for by including the squares of $r_{\mathrm{s}}$ and $r_{\mathrm{d}}$ in (1). If there is a mismatch in radii, the values of $F_{\mathrm{d}}$ must be scaled so that the integral is preserved. For source cells that are completely masked or where data are undefined or "missing", $f_{\mathrm{s}} = 0$. For the most common case when the value in a source cell is representative of the entire cell extent and when there are no missing data, $f_{\mathrm{s}} = 1$ for all grid cells. Note that when remapping conservatively, if $f_{\mathrm{s}} = 1$ for all source cells, then we require that $f_{\mathrm{d}} = 1$ for all destination cells.

A different relationship must be satisfied to preserve a field's global mean (denoted by an overbar):

$$\overline{F_{\mathrm{s}}} = \frac{\sum_i F_{\mathrm{s}i} f_{\mathrm{s}i} A_{\mathrm{s}i}}{\sum_i f_{\mathrm{s}i} A_{\mathrm{s}i}} = \frac{\sum_j F_{\mathrm{d}j} f_{\mathrm{d}j} A_{\mathrm{d}j}}{\sum_j f_{\mathrm{d}j} A_{\mathrm{d}j}} = \overline{F_{\mathrm{d}}} \,. \tag{2}$$

In the next section, we introduce the formulas that apply to remapping data when $f_{\mathrm{s}} = f_{\mathrm{d}} = 1$ everywhere (i.e., when there is no partial or complete masking of any cells). This is followed in section 3 by a description of the more general procedure when a field might be partially masked. In section 4 we provide recipes that should be followed in remapping different types of two-dimensional and three-dimensional fields. That is followed by a brief discussion of how to preserve certain properties when interpolating data in the vertical. In the summary, we include some discussion of when conservative remapping may be inappropriate (or at least inadvisable).

## 2 Remapping without masking

Consider first the simple case in which $f_{\mathrm{s}} = f_{\mathrm{d}} = 1$ for all grid cells. Weights, $w_{ij}$, that preserve the global mean will be used to calculate $F_{\mathrm{d}j}$ through a matrix multiplication:

$$F_{\mathrm{d}j} = \sum_i w_{ij} F_{\mathrm{s}i} \,. \tag{3}$$

These weights would need to be scaled by $r_{\mathrm{s}}^2/r_{\mathrm{d}}^2$, to preserve the global integral, rather than the global mean. Initially, we shall suppose that it is the global mean that should be preserved. At the end of the section 3, we show that the destination field that preserves the global mean can be simply scaled to preserve the integral.

Conceptually, the remapping weights are determined by overlaying the destination grid onto the source grid and calculating what fraction of each source cell overlaps each destination cell. These fractional contributions, here denoted $\omega$, are then multiplied by the ratio of the source cell area to the destination cell area to yield:

$$w_{ij} = \omega_{ij} A_{\mathrm{s}i}/A_{\mathrm{d}j}. \tag{4}$$

Each distinct portion of a source cell must contribute to one and only one destination cell, and all portions must contribute to some destination cell. This implies that within any source cell $i$, the fractional contributions $\omega_{ij}$ must sum to unity:

$$\sum_j \omega_{ij} = 1 \qquad \text{for all } i. \tag{5}$$

This identity can be used to prove that for the case $f_{\mathrm{s}} = f_{\mathrm{d}} = 1$, (3) preserves the global mean. The proof is obtained by substituting (4) into (3), then substituting the result into (2), reversing the order that the summations are performed, and making use of (5). Importantly, a consistent set of areas must be used in evaluating (1), (2), and (5).

Noting that the fractional areas of the source cells contributing to a destination cell must add up to the area of the destination cell, we obtain a second useful identity:

$$\sum_i \omega_{ij} A_{\mathrm{s}i} = A_{\mathrm{d}j} \qquad \text{for all } j. \tag{6}$$

As discussed below, this identity holds only when the fractional contributions, $\omega$, and cell areas, $A$, are consistently defined (i.e., based on the same cell shapes).

Most conservative remapping algorithms are variants of an approach suggested by Dukowicz and Kodis (1987). The most difficult step in remapping conservatively is to devise an algorithm to calculate the fractional contributions, $\omega_{ij}$. Even when efficiently done, the calculation of $\omega_{ij}$ dominates the remapping execution time. In computing $\omega_{ij}$, existing general remapping algorithms (see Appendix A) require the locations of grid cell vertices be specified (for both source and destination grids), and the remapping algorithm must make assumptions as to how the cell vertices are connected. In some remapping algorithms, the cell edges are assumed to run along great circles; in others the cell edges are assumed to coincide with straight lines on a equirectangular projection of the spherical coordinates (as for a regular cartesian longitude by latitude grid, where the latitude cell bounds follow latitude circles, not great circles). Most commonly used packages relied on for remapping conservatively are unable to generate without approximation the weights when cell edges should be constructed under different assumptions (e.g., with cell sides coinciding with great circles on one grid, but not the other). Perhaps the only exception is the YAC ("Yet Another Coupler") interpolation software (see Appendix A).

When cell shapes are misrepresented, the $\omega_{ij}$ values calculated by the remapping algorithm are, in general, only approximations of the true fractional contributions. This means a remapped field can be somewhat distorted, and the errors in the field on the destination grid will generally exceed the formally derived error estimates, especially for coarse grids.

After computing the fractional contributions, $\omega_{ij}$, remapping codes can generate the remapping weights using (4). This ensures that

$$\sum_i F_{si} A_{si} = \sum_j F_{dj} A_{dj}, \tag{7}$$

where the areas are calculated based on the algorithm's assumed cell shapes. These areas may differ somewhat from the true cell areas. If they do, then in general

$$\sum_i F_{si} A_{si} \neq \sum_i F_{si} A_{si}^*, \tag{8}$$

where the asterisk distinguishes a true cell area ($A^*$) from an approximate cell area ($A$) generated by the remapping algorithm. The sum on the right hand side of the equation represents the true integral on the original source grid, but it is the sum on the left hand side that is preserved by (7).

    In order to preserve the true global mean, some packages accept, as an option, user-supplied true cell areas, which are then
used in calculating the remapping weights,

$$w_{ij}^* = \omega_{ij} A_{si}^* / A_{dj}^*, \tag{9}$$

but it should be noted that in this case and in contrast with (6),

$$\sum_i \omega_{ij} A_{si}^* \neq A_{dj}^*. \tag{10}$$

The true fractional areas of the source cells contributing to a destination cell may not add up to the true area of the destination
cell because the true areas may be inconsistent with the shapes of cells assumed in generating the fractional contributions, $\omega_{ij}$.

    Despite this apparent problem, some users may choose to calculate the destination values according to

$$F_{dj}^* = \sum_i w_{ij}^* F_{si}. \tag{11}$$

It follows that

$$\sum_i F_{si} A_{si}^* = \sum_j F_{dj}^* A_{dj}^*. \tag{12}$$

The sum on the left side of the equation represents the true global integral. Thus, when true areas are used in constructing the weights, the remapped field can preserve the true global mean.

    Both alternatives for computing remapping weights, (4) or (9), rely on the same fractional contributions, $\omega_{ij}$, which are based on cell shapes constructed by the remapping algorithm that in some cases are only approximate (e.g., when cells are assumed to be circumscribed by great circles, but in fact have edges following lines of longitude and latitude). With the first
option, the "approximate-area option", cell areas are based on cells bounded by great circle segments, and the weights ($w_{ij}$) are defined by (4). In this case the areas are consistent with the assumed cell perimeters, so (6) holds. With the second option,

the "true-area option", the true grid cell areas must be supplied to the remapping algorithm, and the weights $(w_{ij}^*)$ are defined by (9). In this case the areas may be inconsistent with the assumed cell perimeters used to generate the fractional contributions $(\omega_{ij})$, so there may be a mismatch between the source areas and the destination areas, as noted in (10).

For the purpose of evaluating the relative merits of the two options, we now consider an example of a simple source grid and an idealized temperature field. It should be said upfront that we have selected an example that clearly reveals the consequences of misrepresenting cell shapes and areas. This has dictated that we consider, in the first instance, grid resolutions that are uncommonly coarse. Many climate studies deal with grid cells smaller than a few degrees longitude and latitude, and not tens of degrees, as in the initial example below. It turns out that the size of the remapping errors generally are proportional to the

longitudinal cell widths squared, so that compared to our coarse resolution example, errors would be quite a bit smaller. This should be kept in mind in what follows.

     For our illustrative example, suppose both the source and destination grids are spherical coordinate grids with the same latitude spacing ($15°$) but with the destination grid having half the longitudinal resolution of the source grid ($60°$ vs. $30°$ spacing). Suppose the grids are aligned such that each destination cell completely contains exactly two source cells. The cell

areas are given in Table 1. For the source cells nearest the pole, the true cell areas (with latitude cell bounds following latitude circles) is $0.0178\,r^2$, whereas the approximate cell areas (assuming all cell sides follow great circles) is $0.0171\,r^2$. Thus, there is a 4% error in approximating the cell area. In general these errors increase toward the poles and as grid cell longitudinal width increases. For the destination grid, with twice the angular cell widths, the error quadruples to 17%, whereas halving the cell width shrinks the error to 1%. It can be shown that for small longitudinal cell widths, the fractional error in the approximate

cell areas is proportional to the square of the cell widths.

     In this example, correctly remapping a source field to the destination grid is trivial since each destination value is determined solely by the contributions from the two source cells that alone occupy it. Consider a temperature that varies linearly with latitude and is independent of longitude. Then, the destination field is identical to the source field but with half the longitudinal resolution. The temperature dependence on latitude for the case considered is given by the black line labeled "source grid

(truth)" in Figure 1. Of course when correctly remapped, the destination values should in this case be the same as the source values and also would lie on the black line.

     If, however, we remap this temperature field based on an algorithm that assumes when computing approximate cell areas that the cell boundaries are defined by great circle segments, the destination values will lie on the dashed brown curve in Figure 1. On the other hand, if the algorithm relies on the true cell areas when computing weights (the true-area option), the descrepancy

is much larger, with the resulting destination values given by the dashed blue curve in Figure 1. Neither option correctly remaps the field, but the approximate-area option appears to be far superior to the true-area option.

     Under each option, a global integral can be preserved, according to (12) and (7), but only when true areas are used and the destination field is obtained using (11) can we be certain to preserve the global integral as calculated on the original grid. Since the primary purpose in applying a conservative remapping scheme is to preserve the *true* global mean (i.e., the mean

calculated with true areas), the approximate-area option would seem to be unacceptable. In Figure 1, although the destination

values shown for the approximate-area option appear to nearly coincide with the source values ("truth"), they are in fact systematically underestimated and lead to a global mean temperature 0.45 K cooler than the true mean.

On the other hand, it would seem equally unsatisfactory to adopt the true-area option (dashed blue curve of Figure 1), which produces remapped values at some latitudes differing by more than 30 K from the true values. The true-area option does indeed preserve the true global mean, but the pattern of the destination field can hardly be considered a good representation of the source field. Thus, for different reasons, both options might be considered unacceptable.

It is interesting and somewhat disconcerting to note that with the true-area option, the results of remapping can depend on the units used to express the temperature. The dashed blue curve in Figure 1 shows results when temperature is expressed in kelvins (K). The figure also shows that converting the temperature to degrees Celsius (°C), remapping that field, and then converting it back to kelvins results in considerably smaller errors (crosses in Figure 1). It can be shown that in general, the errors in destination values, when computed using the true-area option, are approximately proportional to the magnitude of the values themselves. Since, on average, we can reduce the mean of the absolute values of a field by removing the global mean before remapping, we can use this strategy to reduce the errors. Our example of converting the temperature units from K to °C was an approximate application of this strategy which reduced the error because $0°$ C is much closer to the mean than $0°$ K. If we were to adopt the more general approach of removing the global mean before remapping, we arrive at the following formula:

$$F_{\mathrm{d}j}^* = \overline{F_{\mathrm{s}}} + \sum_i w_{ij}^* \left( F_{\mathrm{s}i} - \overline{F_{\mathrm{s}}} \right), \tag{13}$$

where here and in what follows an overbar indicates a global mean that must be computed using the true areas, not the approximate areas that remapping algorithms might generate. This variant of the true-area option will subsequently be referred to as the "true-area (centered)" approach, to distinguish it from the "true-area (uncentered)" approach which relies on (11).

An objection to using (13) is that a change in $F_{\mathrm{s}}$ anywhere in the domain will change the mean and thereby impact the destination values everywhere in the domain. One would expect that local remapping should be independent of remote field changes, so (13) would also seem to be less than ideal. It should be noted that if weights generated with the approximate-area option were used in (13), the resulting destination field would be identical to that obtained with (3). This is because for these weights, (6) holds.

Yet another shortcoming of the true-area option is that its application to a spatially uniform source field can result in a destination field with non-zero spatial variance, which is obviously unrealistic. Consider, for example, a source field that everywhere has the value 1. For the grid defined earlier (see Table 1), application of (11) results in a destination field with the same mean (equal to 1), but with area-weighted variance equal to 0.064, and a maximum absolute deviation from the true value of 0.13. These unrealistic variations may in some applications be unacceptable. Algorithms that maintain the uniformity of an originally constant field are said to be "consistent" (e.g., Ullrich and Taylor (2015)), so the true-area option might be described as "not consistent".

For the true-area option, use of (11) or (13) can sometimes result in a destination field with nonphysical values. Consider, for example, a possible result of mapping to a destination grid the ice-free fraction (i.e., the fraction of a grid cell area that is

ice-free). As in the first example above, suppose ice-free fraction is independent of longitude and a linear function of latitude, varying from 0 in the polar-most latitude band to 1 in the latitude band adjacent to the equator. Application of (11) results in a value of 1.06 for the latitude band nearest the equator, and application of (13) results in a value of 1.01. Clearly, the remapping algorithm in both cases yields a nonphysical result with the ice-free fraction exceeding 1 in the tropics. In contrast, the approximate-area option generates destination values that never exceed the maximum or minimum source values. The

approximate-area approach is said to be a monotone method (e.g., Ullrich and Taylor (2015)), whereas the true-area approach is not.

Given the shortcomings of both the centered and uncentered variants of the true-area option, we reconsider the approximate-area option, which relies on the remapping algorithm to construct cell shapes and areas assuming perimeters coincide with great circle segments. The fundamental problem with this approach, as expressed by (8), is that the true global mean (as

calculated on the source grid using true areas) is not generally preserved. In the fields we have examined, we have found that the difference between the true mean and the mean of the field remapped using the approximate area approach is quite small. It seems reasonable, therefore, to simply adjust all values in the field by a uniform amount to correct for the small mismatch in means. For the case considered in Figure 1, 0.45 K can be added to each of the destination grid cell values obtained with the approximate-area option. This straight-forward adjustment eliminates the flaw in the approximate-area option that led us to

discard it originally. This "global adjustment" to the destination field means that local destination field values will be influenced by remote field values. This is undesirable, but as noted earlier, the true-area (centered) approach is similarly impacted. On the other hand, a virtue of the approximate-area option is that unlike the true-area option, a change in units (from, for example, kelvins to °C) does not affect the accuracy of the result. In addition, a source field that has no variations (i.e., is everywhere the same) will map to a destination field that is also constant. Both of these results follow because when applying the approximate-

area option, (6) holds.

Since neither of the cell-area choices offered by remapping codes is without shortcomings, it is worth further examining the characteristics of their errors to determine which approach results in the more realistic representation of the original field. For the temperature field considered earlier, Figure 2 shows the remapping error when approximate cell areas are used (with and without a global mean correction) and when the true cell areas are used with global mean removed and then reapplied following

(13), labeled the "true area (centered) option" in the figure. We limit our discussion for now to all but the dotted curve, which will be discussed in the next section. The correction to preserve the global mean under the approximate-area option by design simply offsets the curve by the same amount everywhere. With this correction the "approximate-area (corrected uniformly) option" (as it will be referred to subsequently) has the smallest root-mean-square error (RMS error, calculated with grid cell area weighting). The RMS errors are: 0.15 K for the approximate-area (corrected uniformly) option, 0.48 K for the

approximate-area (uncorrected) option, and 0.63 K for the true-area option (centered). In this example, although the correct global mean is preserved under both the true-area (centered) option and the approximate-area (corrected uniformly) option, the second option results in an RMS error a factor of 4 smaller than the first option.

In Figure 3 we consider for a spherical coordinate (cartesian longitude by latitude) grid how the remapping errors depend on grid cell size. The source data were taken from Figure 1 with values independent of longitude location and 15° latitude spacing.

Source grids with longitudinal resolution from $0.5°$ to $30°$ were considered, and in each case the destination grid had half the longitudinal resolution of the source. As discussed above, the errors arise because of inaccuracies in the representation of grid cell shapes by the remapping algorithm (which assumes the cell latitude bounds follow great circles, not latitude circles). It is not surprising then that the finer the resolution, the smaller the errors (because for small grid cells, the great circles deviate very little from latitude circles). Figure 3 shows that under all options the RMS error is proportional to the square of the grid cell's longitude width and that for any given resolution, the approximate-area option error is about 1/4 the size of the true-area (centered) option error and more than two orders of magnitude smaller than the error in uncentered true-area option. From another perspective, compared to the true-area (centered) option, the approximate-area option can with equal accuracy handle grids that are twice as coarse.

Figure 3 shows that for grid resolutions typically used in climate research (a few degrees longitude width and finer), the RMS errors would be more than two orders of magnitude smaller than the errors at the coarsest grid resolution considered ($60°$). For the approximate area (uncorrected) option, the RMS error is reduced from about 0.48 K at the coarsest resolution to about 0.002 K at $4°$. Similarly, the global mean correction needed at these two resolutions is 0.45 K and 0.002 K, respectively. For some studies an error of half a degree would be of concern, but an error of few thousandths of a degree might be considered acceptable. In some cases, then, a remapped field that does not quite preserve the true global mean might be considered adequate and not require correction, but this will depend on what kind of use is being made of the data.

Recall that when there are some physical limits on a variable (e.g., a fraction confined to the interval 0 to 1), remapping algorithms may not respect those limits. Although with the approximate-area option, the remapping step ensures that all destination values will be within the maximum and minimum values of the source values, the correction to the mean required when applying that option can sometimes push values outside the limits. This issue can be addressed with a refined correction, which will be described in a part of the next section.

## 3  Remapping of partially or fully masked cells

We now consider the more general procedure for conservative remapping when there might be undefined elements in the source array (e.g., missing or masked elements) or when grid cell values might be defined for only a fractional portion of the source cell (for example only over the land portion of a cell). For this purpose, we will adopt and generalize the form of the approximate-area option because, as discussed above, it was found to be more accurate than the true-area options and because with this option we can simplify some subsequent formulas using (6). Moreover, where the field is constant on the source grid, the approximate-area option, unlike the true-area option, does not introduce unrealistic spatial variations in a region.

The key to handling data that may represent conditions on only a portion of each grid cell is to specify for each cell the "unmasked" fraction, and when remapping is performed, generate the appropriate destination unmasked fractions. Although sometimes the source grid unmasked fractions are binary (either 0 or 1) and might be inferred from special bit strings indicating "missing" data, if the data are remapped, the unmasked fractions will in general no longer be binary, and thus information will

be lost unless the unmasked fractions are carried as an additional field along with the data field. The key to general remapping then is to carefully account for the unmasked fractions and to ensure that they are consistently defined on the destination grid.

Generalizing (3) to account for fully or partially masked fields requires modification of the weights defined by (4). This is done by replacing in (4) the ratio of areas by the ratio of *unmasked* portions of the cell areas. Alternatively and equivalently, we can keep the original, unmasked definition of weights and explicitly include the unmasked fractions in (3), which then becomes

$$\hat{F}_{\mathrm{d}j} = \sum_i w_{ij}(f_{\mathrm{s}i}/\hat{f}_{\mathrm{d}j})F_{\mathrm{s}i}\,, \tag{14}$$

where the 'hats' atop $F_{\mathrm{d}j}$ and $f_{\mathrm{d}j}$ indicate that these are preliminary estimates of destination values which might need subsequent correction to preserve their true global means. Note that the source unmasked fractions, $f_{\mathrm{s}i}$, must be set to 0 wherever there are missing data. When this is done, missing data need not be treated specially because the missing value will invariably get multiplied by $f_{\mathrm{s}i}$, yielding 0 for the product. This ensures that missing values have no impact on the remapped fields.

For some applications, destination fractions may have been imposed as part of the definition of the destination grid. For the purposes of remapping a field, however, it is essential that the destination fractions in (14) be defined such that

$$\hat{f}_{\mathrm{d}j} = \sum_i w_{ij} f_{\mathrm{s}i}\,. \tag{15}$$

The remapping formula (14) can then be written

$$\hat{F}_{\mathrm{d}j} = \frac{\sum_i w_{ij} f_{\mathrm{s}i} F_{\mathrm{s}i}}{\sum_i w_{ij} f_{\mathrm{s}i}}\,. \tag{16}$$

Thus, the destination value is simply a weighted mean of the contributing grid cell values. This ensures that the destination value will not lie outside the maximum and minimum values of the contributing cells. This further implies that if all source cells contributing to some destination cell have the same value, the destination cell will also be assigned that value. Note that if in (16) the sum in the denominator is zero, then the destination value should be designated as missing. Existing remapping packages presumably have provided options for calculating the destination values using (16), but some may require $f_{si}$ to be a binary mask (unmasked or fully masked) rather than allowing for partial masking.

As shown earlier, use of approximate areas in computing the weights in (16) does not in general preserve the true global mean of $F$. As in the simpler case, a global adjustment to the $\hat{F}_{\mathrm{d}j}$ values must be applied, but here we allow the correction to vary spatially,

$$F_{\mathrm{d}j} = \hat{F}_{\mathrm{d}j} - \frac{\gamma_j}{\overline{\gamma}}\left(\overline{\hat{F}_{\mathrm{d}}} - \overline{F_{\mathrm{s}}}\right), \tag{17}$$

where $\gamma_j/\overline{\gamma}$ is a spatially varying correction coefficient, and the global mean of $\gamma$ in the denominator above ensures $\overline{F_{\mathrm{d}}} = \overline{F_{\mathrm{s}}}$. With masking, the global mean quantities (indicated by overbars) must be computed with area weights proportional to only the *unmasked* area of grid cells, For example,

$$\overline{\hat{F}_{\mathrm{d}}} = \frac{\sum_j \hat{F}_{\mathrm{d}j} f_{\mathrm{d}j} A_{\mathrm{d}j}^*}{\sum_j f_{\mathrm{d}j} A_{\mathrm{d}j}^*}\,. \tag{18}$$

Note that in this formula, the unmasked fraction, $f_d$, is not necessarily identical to the unmasked fraction, $\hat{f}_d$, which appears in (14). Sometimes, for example, the remapped data must conform to a destination grid with an imposed masked region. In that case, the already defined fractions, $f_d$, can be used in (18). This, however, could result in some destination field values calculated with (16) being masked. With those values no longer contributing to the destination field, the correction to the mean given by (17) must compensate, and this will alter the destination values, $F_{dj}$, globally, not just locally. It is therefore advisable to assign destination masked fractions consistent with (15) and avoid imposing externally defined destination masked fractions.

In the simplest case, the correction coefficient in (17) is set to 1 for all $j$, which adjusts each cell value by the same amount everywhere (i.e., by $\overline{\Delta F} = \overline{\hat{F}_d} - \overline{F_s}$). This can, however, lead to nonphysical results. Suppose, for example, that a positive definite quantity (such as the liquid water content of air) were mapped to a target grid using (16) and that the resulting global mean were greater than $\overline{F_s}$. In this case, any cell with $\hat{F}_{dj} = 0$ would, after a simple adjustment with $\gamma_j = 1$, become negative, which must be ruled out on physical grounds.

More generally, a uniform adjustment of the destination field may result in values that lie outside the range of source values. Returning to our earlier example, we see in Figure 2 that a uniform correction to the temperature field of 0.45 K results in a positive 0.1 K error in the equator-most cell, and the remapped temperature there is warmer (by 0.1 K) than the warmest temperature found in the original field (290 K). Thus, the remapped field, which before correction was monotonic, is no longer so.

To remedy this undesirable consequence of a uniform correction, $\gamma_j$ should vary such that the original maximum and minimum temperatures are not exceeded. A number of methods have been developed to ensure the monotonicity of remapped fields (e.g., Zerroukat et al., 2005; Schneider et al., 2018; Bradley et al., 2019; Lauritzen and Nair, 2008). Here as alternatives to some of the more sophisticated correction methods, we offer two options that would be easy to include in remapping procedures.

We consider first the case of a positive definite field, such as the concentration of a trace species. In this case we suggest that rather than apply a uniform increase or decrease in values, the same *fractional* correction be applied across the entire field. This is achieved by setting the coefficient in (17) to the local field value itself, $\gamma_j = \hat{F}_{dj}$. With this, (17) reduces to:

$$F_{dj} = \hat{F}_{dj} \frac{\overline{F_s}}{\overline{\hat{F}_d}}. \tag{19}$$

This scaling ensures that when the concentration varies over orders of magnitude, most of the correction needed to preserve the global mean will be accomplished where concentrations are relatively high. When considering smoke concentration, for example, a correction to the total mass of smoke would be made primarily in the smoke plume, not in the smoke-free regions. It should be noted, however, that while (19) ensures concentrations are nowhere negative, the maximum concentration in the remapped field might exceed the maximum value in the original field.

There are some cases where certain limits must be strictly enforced. As an example, the fraction of each grid cell covered with sea ice must never be negative or exceed 1. To preserve the global mean while respecting these limits, we can apply (17) with $\gamma_j$ defined as

$$\gamma_j = (\hat{F}_{dj} - F_{\min})^\mu (F_{\max} - \hat{F}_{dj})^\mu, \tag{20}$$

where $F_\mathrm{min}$ and $F_\mathrm{max}$ represent the imposed upper and lower limits of the field. For a fraction like sea ice these would be set to 0 and 1. The same form for $\gamma$ applies when we require monotone remapping, but the limits in this case would be set to the source field's minimum and maximum values. With this approach, values near the maximum and minimum values are adjusted by a smaller amount than values nearer the middle of the range of values. Figure 4, which applies to the temperature field considered earlier, shows how the distribution of the error is determined by $\mu$. A small value of $\mu$ will distribute the correction fairly evenly throughout the range except near the limits. A large value of $\mu$ saddles the middle range with most of the adjustment. No matter what the value of $\mu$, there is no correction to values already at the maximum or minimum.

Ideally, we might choose to distribute a needed correction according to where the grid cell shape misrepresentations are largest (and where the local conservation errors are likely largest). There is, however, no easy way to do this. Instead we choose to simply distribute the correction according to (17) with $\gamma$ defined in (20) and with $\mu$ chosen such that a relatively even correction is applied everywhere without pushing the values of any cell beyond the imposed maximum and minimum limits. Consistent with this intent, we describe in Appendix B a procedure whereby the value of $\mu$ can be found that maximally "flattens" the $\gamma$ curve while ensuring that all field values remain within the imposed limits. This will be referred to subsequently as the "approximate area (corrected and monotone)" option.

In the example considered above, the temperature in the cell adjacent to the equator would, as already noted, exceed by 0.1 K the maximum temperature found in the original field if a uniform correction were applied. To prevent this, we distribute the correction needed to preserve the true global mean according to (20), as just discussed. The value of $\mu$ appearing in the correction coefficient is 0.161, as determined by the procedure described in Appendix B. The dotted curve in Figure 2 shows the result of applying this nonuniform correction. In addition to eliminating the artificial maximum resulting from a uniform correction, the nonuniform correction slightly reduces the RMS error in the remapped temperature field. Figure 3 shows that at all resolutions, a nonuniform correction (with $\mu$ values obtained as described in Appendix B and listed in Table 2) produces a temperature field with the smallest errors of all options considered.

In the case of coarse resolution (with longitude widths $\geq 30°$), the uncorrected global mean of the remapped field is 0.45 K cooler than the true global mean. For finer resolution grids, the magnitude of the correction is considerably smaller: when, for example, the same temperature field is mapped from a source grid with cell widths of 4 degrees longitude to a grid with widths of 8 degrees, the global mean is less than 0.01 K cooler than the true mean. When it is *not* essential to preserve the true mean, one might choose to accept such a small error in global mean in order to skip the correction procedure described above.

A final adjustment to $F_{\mathrm{d}j}$ may be needed if the objective is to preserve the global *integral* of a field, rather than the global *mean*. (To conserve energy in a coupled climate model, for example, it is the surface heat flux between the atmospheric component and the ocean component that might need to be mapped from one grid to another, and it is the total flux which must be preserved.) Comparison of (1) and (2) indicates that to preserve the integral, the values of $F_{\mathrm{d}j}$ obtained using (17), which preserves the mean, should be scaled by the ratio of the global unmasked source area to the unmasked destination area:

$$c = \frac{r_\mathrm{s}^2 \sum_i f_{\mathrm{s}i} A_{\mathrm{s}i}^*}{r_\mathrm{d}^2 \sum_j f_{\mathrm{d}j} A_{\mathrm{d}j}^*}. \tag{21}$$

If the destination unmasked grid-cell fractions have been defined such that the global mean unmasked fraction is preserved, the sums in the numerator and denominator of (21) are the same, and the formula simplifies to $c = (r_\mathrm{s}/r_\mathrm{d})^2$. When this is

true and if $r_\mathrm{d} = r_\mathrm{s}$, the same destination field, without scaling, preserves both the mean and the integral. Note, however, if the destination unmasked fractions have been calculated with (15) and they have *not* been corrected to preserve the global mean unmasked fraction, then $c$ must be calculated with (21).

## 4   Recipes for regridding

Some conservative remapping packages (see Appendix A) may not be designed or may not clearly document how to handle the

most general cases considered here (e.g., fields with missing values or grid cells that are partially masked). Those codes may nevertheless be relied on to provide weights defined by (4). For a given source and destination pair of grids, these weights can be calculated once, and then used to remap any field from the source grid to the destination grid, even fields that are partially masked.

A step-by-step procedure for mapping variables conservatively is provided here.

1. Obtain from a remapping package the weights ($w_{ij}$) that apply when there is no masking or fractional weighting. Accept that these weights might be based on the algorithm's sometimes inaccurate reconstruction of grid cell shapes.

2. Check that for all destination cells the weights satisfy (6), which with (4) can be rewritten:

$$\sum_i w_{ij} = 1 \qquad \text{for all } j. \tag{22}$$

(It is prudent to include this step but not necessary if the weights returned by the remapping package are known to satisfy

this requirement.) If (22) is satisfied, the remapping algorithm will be "consistent" in the sense that a spatially constant source field will remain spatially uniform on the destination grid.

3. Assign or calculate the source grid's unmasked fractions, $f_{\mathrm{s}j}$.

    – If a source value is meant to represent conditions over the entire cell extent, set $f_{\mathrm{s}i} = 1$ for the cell.

    – If unmasked fractions have been assigned to source cells prior to remapping, the pre-assigned values should be

assigned to $f_\mathrm{s}$.

    – Wherever source cell data are missing, reset the unmasked fraction to 0 ($f_{\mathrm{s}i} = 0$).

4. Assign or calculate the destination grid's unmasked fractions, $f_{\mathrm{d}j}$.

    – If unmasked fractions have been assigned to destination cells prior to initiating the remapping procedure, $f_\mathrm{d}$ may be set to the pre-assigned values. As noted in the previous section, however, when destination masked values are

not consistent with (15), the destination field, $F_\mathrm{d}$, will be impacted everywhere, so when possible, avoid applying external destination masked fractions different from $\hat{f}_\mathrm{d}$.

- If unmasked fractions have not been pre-assigned, generate the fractions with (15). When necessary and desirable, correct the values of $\hat{f}$ to preserve the global mean fraction by applying formulas analogous to (17) and (20).

$$f_{\mathrm{d}j} = \hat{f}_{\mathrm{d}j} - \frac{\gamma_j}{\bar{\gamma}}\left(\overline{\hat{f}_{\mathrm{d}}} - \overline{f_{\mathrm{s}}}\right) \tag{23}$$

and

$$\gamma_j = \hat{f}_{\mathrm{d}j}^{\mu}(1 - \hat{f}_{\mathrm{d}j})^{\mu}. \tag{24}$$

This step ensures that the same destination field will preserve both the global integral and mean under the conditions discussed following (21). The value of $\mu$ should be determined following the procedure described in Appendix B. The global means of $\hat{f}_{\mathrm{d}}$ and $f_{\mathrm{s}}$ in the first equation above must be calculated weighted by the full true areas ($A_{\mathrm{d}j}^*$ and $A_{\mathrm{s}i}^*$).

5. Use (16) to calculate the preliminary destination values, $\hat{F}_{\mathrm{d}j}$. For any cell where the denominator in (16) is 0, designate the destination value as "missing".

6. When necessary and desirable, correct the destination values, $\hat{F}_{\mathrm{d}j}$, to preserve the true global mean and obtain the final destination field, $F_{\mathrm{d}}$. In the next two sub-steps, when calculating (17), global means of $F_{\mathrm{s}i}$ and $\hat{F}_{\mathrm{d}j}$ should be weighted by $f_{\mathrm{s}i}A_{\mathrm{s}i}^*$ and $f_{\mathrm{d}j}A_{\mathrm{d}j}^*$, respectively.

- Initially, attempt to impose a uniform correction to all values by applying (17) with $\gamma_j = 1$ for all $j$. If none of the resulting $F_{\mathrm{d}j}$ values have been shifted outside the extremes of the source field, consider the correction acceptable; otherwise recover the original $F_d$ field and proceed.

- If the uniform adjustment is unacceptable, apply (17) with $\gamma$ defined by (20) and $\mu$ determined following the procedure outlined in Appendix B.

Note that no correction of $\hat{F}_{\mathrm{d}j}$ is needed if two conditions are met: 1) the remapping algorithm correctly represents both the source and the destination grid cell shapes (in which case the fractional contributions, $\omega$, and cell areas will both have been correctly determined), and 2) the unmasked fractions on the destination grid are defined by (15) and have not been corrected to preserve the global mean fraction.

In what follows, the above recipe will be referred to as the "standard procedure". The weights, $w_{ij}$, only depend on the source grid and destination grid, so a single set of weights can be generated that can be applied in remapping any field. It should be noted that the approximate areas calculated by the remapping algorithm are of no interest once the mapping weights have been generated. In the subsequent mapping of a variable from the source grid to the destination grid, only the true areas of cells may be needed.

Care must be taken when the standard procedure for remapping is applied to a variable representing conditions within layers of the atmosphere or ocean to ensure that mass-weighted means are preserved (as opposed to the usual area-weighted means).

Additional complications might be encountered when a variable represents the ratio of two quantities (e.g., specific humidity is the ratio of the mass of water vapor to the mass of air), where, rather than preserving the global mean ratio, it is better to preserve the two quantities themselves. The following guidelines may be helpful in treating these possibly troublesome variables.

(a) To remap a quantity representing a *grid cell area fraction* (e.g., cloud fraction, sea ice fraction, land fraction), the destination fractions should be calculated in the same way as unmasked fractions were calculated in step 4 above.

(b) To conserve a *vertical flux of energy* through a surface, the flux must be expressed as a flux per unit area ("flux density" with units of, for example, W/m$^2$, not W). Then the standard procedure is followed to remap the flux density to the destination grid where it is scaled by $c$, as defined by (21).

(c) To remap the *albedo* (reflected radiation divided by incident radiation), which is undefined when the incident radiation is zero, it is best to conservatively remap the incident and reflected radiation flux densities (commonly termed "radiative fluxes") and then form their quotient, rather than directly remapping the albedo. Destination values should be considered "missing" (undefined) where the remapped incident radiation is 0.

(d) There are applications where the *total volume* of a space (which might be partially masked) should be preserved. For grids constructed with height (or depth) used as a vertical coordinate, this can be achieved by calculating appropriate cell thicknesses on the destination grid. The standard procedure above is followed, applied to cell thickness. The resulting values must be scaled by $c$, as defined by (21). The unmasked portion of a destination cell volume is the product of the remapped cell depth, the destination cell fraction that is unmasked ($f_{\mathrm{d}j}$), and the true destination cell area ($A_{\mathrm{d}j}^*$).

(e) For most 3-d quantities, remapping should preserve the mass-weighted mean, rather than the area-weighted mean. Prior to remapping such variables, the *mass field* must be remapped conservatively. In order to preserve the total mass within a layer, the mass per unit area ($M$) of destination cells can be obtained following the standard remapping procedure. We consider two cases: models for which the bounds on layers can be expressed as a pressure and models for which the bounds on layers can be expressed as a distance.

When the layer pressure thicknesses can be determined, the mass per unit area in the layer is $M = \Delta p / g$ (where $\Delta p$ is the pressure thickness of the layer, which may vary with longitude and latitude, and $g$ is the acceleration due to gravity). Preserving the mass within a layer is equivalent to preserving the pressure thickness of the layer. This is achieved following the standard remapping procedure with $F_{\mathrm{s}i} = \Delta p_{\mathrm{s}i}$ and scaling the result by $c$, as defined by (21).

When the layer thicknesses can be determined, the layer mass per unit area is equal to the product of cell density ($\rho$) and layer thickness ($\Delta z$), so that standard procedure is applied to $\rho \Delta z_{\mathrm{s}i}$. Again, the result is scaled by $c$, as defined by (21). By this procedure we preserve global mass, but we should also like to define density and layer thickness on the destination grid such that their product is $M_{\mathrm{d}j}$. For models with a uniform source-grid layer thickness, $\Delta z$, it makes sense for the source-grid thickness to carry over to the destination grid. Then the density is given by $\rho_{\mathrm{d}j} = M_{\mathrm{d}j}/\Delta z$. If,

instead, density is uniform in a source grid layer, then $\Delta z_{\mathrm{d}j} = M_{\mathrm{d}j}/\rho$. If, however, both density and thickness vary on the source grid, then one can choose whether to preserve the global mean layer thickness or the global mean density. One of these fields can be remapped, preserving its mean, and then the other calculated by dividing $M_{\mathrm{d}j}$ by the first field.

(f) Once the mass per unit area is obtained for each destination grid cell, as just described in (e) above, the formula for preserving *mass-weighted integrals or means* can be derived. For example, to remap temperature ($T$) in a layer such that the total internal energy is conserved within a layer, remap the temperature, weighted by each grid cell's mass per unit area, and then divide by the cell mass per unit area on the destination grid ($M_{\mathrm{d}j}$). For a pressure coordinate model with a layer thickness ($\Delta p_{\mathrm{s}i}$) that depends on location, the mass-weighted temperature $U = T\Delta p$, which is proportional to internal energy per unit area, is mapped to the destination grid following the standard procedure. The result, $U_{\mathrm{d}j}$, is then divided by the pressure thickness on the destination grid (defined, as described in (e), such that the global mass is preserved), yielding the temperature field consistent with conservation of internal energy: $T_{\mathrm{d}j} = U_{\mathrm{d}j}/\Delta p_{\mathrm{d}j}$. For a height or depth coordinate model, $\Delta p$ is replaced by $\rho\Delta z$ in the above formulas, with care taken to preserve the global mass in the layer. Note that for layers of uniform mass thickness (either constant $\Delta p$ or constant $\rho\Delta z$), there is no need to consider mass, and instead, the simpler procedure described in (b) can be applied directly without regard to layer thickness.

(g) The amount of a substance in a layer of the atmosphere or ocean is often expressed as a ratio. To remap quantities of this kind, separately remap the quantities represented by the numerator and denominator and then form their ratio, as in the following examples:

– for *specific humidity*, $q$ (mass of water vapor divided by mass of air containing the water), preserve separately the mass of water vapor and the mass of the air. First conservatively remap the water vapor mass per unit area ($q_{\mathrm{s}i}M_{\mathrm{s}i}$). Then remap the mass of air per unit area ($M_{\mathrm{s}i}$), as described in (e) above. Finally, form the ratio of the two remapped fields to obtain the specific humidity on the destination grid. A similar procedure can be applied in remapping any mass fraction.

– for *water vapor mixing ratio* (mass of water vapor divided by mass of dry air), preserve separately the mass of the water vapor and the mass of dry air. A similar procedure can be applied in remapping any mass mixing ratio.

– for *number concentration* (number of particles divided by volume), preserve separately the number of particles and the volume. Then form their ratio.

– for *mass concentration* (mass of substance divided by total volume of mixture), preserve separately the mass of the substance and the volume. Then form their ratio.

– for *mole concentration* (number of moles per unit volume of, for example, a chemical species in the atmosphere or ocean), preserve separately the moles of the substance and the volume. Then form their ratio.

– for *volume mixing ratio* (number of moles of a constituent divided by number of moles of all constituents combined; sometimes referred to as mole fraction), preserve separately the number of moles of the constituent of interest and the number of moles of all constituents combined. Then form their ratio.

(h) In remapping *relative humidity* (mixing ratio divided by saturation mixing ratio), one might want to preserve the relationship between the remapped mixing ratio and the remapped temperature used to define saturation mixing ratio. That

is, one might want the relative humidity on the destination grid to be defined by the ratio of a conservatively remapped mixing ratio divided by a saturation mixing ratio based on a conservatively remapped temperature.

## 5   Interpolating conservatively in the vertical

When remapping a 3-d field both vertically and horizontally, the vertical dimension must be handled carefully to preserve a global mass-weighted integral. When coupling component models (e.g., an atmospheric dynamical core with an atmospheric

chemistry module) specialized handling might be required, but for the purposes of remapping model results, it might be satisfactory to treat the horizontal and vertical dimensions sequentially. We consider here the specific case of first interpolating from a model's native vertical grid to surfaces of constant mass per unit area and then remapping horizontally.

Compared with the generation of weights needed to remap conservatively in the horizontal, it is much easier to define the weights that will preserve integrals in the vertical. This is because the overlap of source and destination grid cells in the vertical

is one dimensional, and only the cell thicknesses must be considered (not their shapes). For data stored on native model levels, bounds defining the vertical extent of each grid cell are an essential component of the grid definition and should be known. Furthermore, the pressures associated with those interfaces should be derivable. Then the mass per unit area contained within the upper and lower bound of a layer can be calculated by dividing the pressure difference across the layer by $g$. The weights can then be obtained by overlaying the pressure bounds from the native grid onto the destination grid bounds and determining the

fraction of each source cell that lies within the vertical extent of each destination cell. These fractions are used to remap the data in the vertical through matrix multiplication. A difference from horizontal remapping is that the weights are not uniform across the other dimensions of the data; they can vary from one location to another and may evolve over time (e.g., in the atmospheric surface layer where surface pressure may vary in time). It is therefore not possible to calculate the weights once and for all as is done in horizontal remapping. Fortunately, calculating the weights for interpolating in the vertical is computationally much

less demanding than in the horizontal, so remapping 3-d fields remains practical.

Once the vertical integration has been completed, conservative remapping of each layer can proceed following the standard procedure summarized in the previous section.

## 6   Summary and concluding remarks

Most conservative remapping packages (see Appendix A) generate mapping weights based on grid cells that for certain grids

might differ slightly in shape from the true cell shapes. Typically, a remapping algorithm will construct cell polygons with

edges that follow great circles and then use these to determine cell areas and mapping weights. On the other hand, many models and analysis grids are constructed on spherical grids with grid cell bounds that follow lines of constant longitude and lines of constant latitude (not great circles). If data are mapped from or to a grid of this kind, the remapping algorithms can fail to preserve the true global mean or integral of a field. The algorithms preserve instead a global mean based on its approximate representation of cell shapes and areas, which generally differs from the true mean. Other packages may assume cell shapes are defined by bounds coinciding with straight lines on a equirectangular projection, and then the cells shapes for the increasingly popular cubed-sphere grid (among many others), which follow great circles, are not accurately represented.

Errors in conservation may especially matter when gauging whether a model, having reached equilibrium, is conserving energy. If the global mean net top of the atmosphere energy flux is in fact zero, as evaluated based on the original grid and correct cell areas, remapping that data and calculating the mean on a new grid could lead to a different conclusion. Similarly, when the mass of a trace species is not preserved, it is impossible to accurately track its changes and possibly determine what the causes of those changes are.

Another limitation of many remapping packages is that although they may be able to treat gridded data where a binary mask applies (e.g., screening regions of missing data or limiting analysis to the ocean or land regions alone), not all are designed to conveniently handle data values that are representative of only a portion of a grid cell (i.e., are partially unmasked). Moreover, often the easiest option offered for handling such cases is to perform the computationally intensive recalculation of weights each time a new mask is imposed.

Here we have explained how users can apply the weights provided by remapping packages, while avoiding or correcting their limitations. For a given pair of source and destination grids, the remapping weights need only be calculated once; the weights are independent of any full or partial masking of the source data. Each destination field can then be calculated via very sparse-matrix multiplications. The recipes appearing in section 4 provide step-by-step instructions on how to handle various cases. These recipes apply even when the remapping algorithm has correctly represented the shapes and areas of grid cells; when that is true, the steps involving correction of the mean can be skipped.

Conservative remapping of the kind considered here must always operate on variables that are independent of the cell's area. For example, rather than remap the area of snow cover in grid cells, the areas first must be converted to fractions, which can be conservatively remapped and then converted back to areas. Most variables reported from models are intensive, so such conversions are rarely necessary.

Conservative mapping is obviously required if it is important to preserve the global integrals (or means) of a field. When this is not essential, other methods of interpolating data to a destination grid may lead to a more physically consistent and realistic looking result. Consider, for example, the geopotential height and wind fields carried on a relatively coarse source grid. If these fields were mapped conservatively to a much finer resolution grid, box-fill contour plots of the resultant fields would look like slightly blurred versions of the box-fill plots of the original fields (often referred to as the mesh-imprinting phenomena). The sub-cells wholly contained within a given source cell would all share the same value; there would be no variation except for the relatively few cells at the borders of the original source cells. Thus, within the confines of each original source cell, the geopotential height and winds would be constant, and at the borders of original source cells, there would be large gradients.

With non-zero wind values but no geopotential gradients within the confines of the original cell, the geostrophic balance generally prevailing outside the tropics would be upset. In general, when mapping from a coarse to a fine grid, a second-order conservative scheme (e.g., Kritsikis et al., 2017) or simple bilinear interpolation should lead to more realistic gradients and more realistic looking results than first-order conservative remapping.

By way of simple examples, we have shown that certain approaches to applying weights generated with commonly used remapping packages can lead to substantial errors even if the true global mean is preserved. The "standard procedure" recommended here avoids some of the problems, but for some grids it can can include a typically small, but perhaps non-negligible, correction to preserve the global mean. For some applications the correction might not be considered large enough to warrant applying it. In this regard it should be remembered that the largest remapping errors illustrated by the simple examples consid-

ered above were associated with very coarse grids. Errors are much smaller and perhaps could be considered insignificant for grids of a more usual, finer resolution, since the errors scale with the square of the grid cell longitude width.

     We have shown that with available remapping packages, careful application of the remapping weights, unmasked fractions, and (when needed) the application of a correction can result in reasonably accurate results with the true global integrals or means of interest preserved. If cell shapes were invariably reconstructed correctly by the remapping algorithms, no correc-

tion would be needed to preserve the global mean and the standard remapping procedure could be simplified. This would seem to provide strong motivation for augmenting remapping packages with the option to construct correctly the commonly encountered longitude by latitude grids with cell edges conforming to the true grid cell shapes.

*Code availability.* The calculations performed in support of the research reported here were based on straight-forward application of the procedures fully described in the manuscript. The calculations could have been performed using an electronic calculator but for this study

were obtained with the assistance of Excel for Mac (version 16.78). The Excel spreadsheet does not conveniently expose the code that produces its numbers, so the best way to reproduce the results reported in this paper is to independently apply the simple formulas to the artificial data described in the text. The spreadsheet is not general enough to treat cases different from the one reported on in this paper, rendering it of little value beyond the current study.

## Appendix A: Remapping packages

The focus here has been how to accurately preserve the global mean of a field when it is remapped to a different grid. There are, of course, other criteria for judging the relative merits of a remapping scheme unrelated to conservation. Valcke et al. (2022) and Mahadevan et al. (2022) define a variety metrics for comparing the properties of remapping algorithms and then apply them to evaluate a number of different remapping packages. We list below a somewhat longer list of packages for generating weights that then can be applied in the second step of the remapping procedure as fully described above:

– C-Coupler2: This package was developed for use in coupling components of climate models (Liu et al., 2018) and includes a conservative remapping capability.

– Climate Data Operators (CDO): This package, designed to manipulate and analyze climate and weather prediction data includes a conservative remapping option that is based on the YAC package (see below). Documentation is available at https://code.mpimet.mpg.de/projects/cdo/wiki.

– Earth System Modeling Framework (ESMF) Regrid Weight Generator (ERWG): This library contains a number of remapping methods useful in the analysis of climate data, including a conservative option. Cell vertices are connected following great circles. Documentation is available at https://earthsystemmodeling.org/regrid/.

– Mesh-Oriented datABase (MOAB): This library (Mahadevan et al., 2020) provides support for coupling model components. It can remap fields conservatively using the weights generated by TempestRegrid.

– netCDF Operators (NCO): This toolkit manipulates and analyzes data of interest to the geophysical community and includes three options for creating remapping weights: TempestRemap, ESMF (ERWG), and NCO's own conservative weight generator, which assumes cell shapes are defined by great circles. Documentation and guidance is available at https://sourceforge.net/projects/nco/, https://nco.sourceforge.net/nco.pdf, and https://acme-climate.atlassian.net/wiki/spaces/DOC/pages/754286611/Regridding+E3SM+Data+with+ncremap.

– OASIS: This software was developed for coupling components of climate models (Craig et al., 2017; Valcke and Piacentini, 2019; Jonville and Valcke, 2019). In its latest version it relies on remapping weights generated offline (with, for example, ESMF or SCRIP). Documentation is available at https://oasis.cerfacs.fr/en/documentation.

– Spherical Coordinate Remapping and Interpolation Package (SCRIP): This is the first library to implement the Jones (1999) approach to remap conservatively (see https://github.com/SCRIP-Project). It assumes the cell sides are in general straight lines on an equirectangular projection so that for the special case of a cell side connecting two points at the same latitude, the side coincides with a latitude circle, not a great circle. Note that SCRIP offers an option to construct cells located very near the poles using Lambert projections.

– TempestRemap: This is a conservative, consistent and monotone remapping package for arbitrary grid geometry with support for finite volumes and finite elements (see Ullrich and Taylor, 2015; Ullrich et al., 2016). The package assumes that cell sides coincide with great circles.

– XML-IO-Server (XIOS): This open source library handles I/O management in climate codes, and it includes a remapping capability. In constructing grid cells, it connects the cell vertices following great circles. In some cases a cell side that begins and ends at the same latitude and spans a large longitude range can be subdivided into multiple short segments (with each segment following a great circle). This results in a side that more nearly follows a latitude circle, and for some grids this can improve global conservation. Documentation is available at http://forge.ipsl.jussieu.fr/ioserver.

– Yet Another Coupler (YAC): A conservative remapping algorithm is included in this climate model component coupler (Hanke et al. (2016); see also https://dkrz-sw.gitlab-pages.dkrz.de/yac). Depending on the grid, cells are constructed with

sides coinciding with great circles or following lines of constant latitude or longitude. This package may therefore be unique in being able to accurately construct grid cell shapes and areas for most grids used in climate models.

## Appendix B: A procedure for determining $\mu$

The value of $\mu$, which appears in (20), is chosen such that the corrections needed to preserve the global mean are distributed as evenly as possible across all cells without violating the maximum and minimum limits imposed on the field. For monotone remapping, the maximum and minimum values are taken as the maximum and minimum values of the original field. When monotonicity is not required but when a field has inherent limits (e.g., the sea ice fraction limits of 0 and 1), those limits define the maximum and minimum values.

Sometimes it is possible to uniformly apply the global mean correction, $\overline{\Delta F}$, to all cells. If the limits of the original field should be respected, we can apply a uniform correction to all remapped cell values only if in all cells:

$$
|\overline{\Delta F}| < F_{\max} - \hat{F}_{\mathrm{d}j} \quad \text{if } \overline{\Delta \mathrm{F}} > 0
$$
$$
|\overline{\Delta F}| < \hat{F}_{\mathrm{d}j} - F_{\min} \quad \text{if } \overline{\Delta \mathrm{F}} < 0. \tag{B1}
$$

When these conditions hold for all $j$, $\mu$ in (20) is specified to be 0, and in (17), $\gamma_j = 1$ for all $j$.

When the conditions of (B1) are not met, then a uniform correction cannot be applied and instead $\mu$ in (20) is chosen such that the correction is distributed across the field as evenly as possible without violating the constraint on maximum and minimum values. As a first step, we define a normalized and centered variable representing the destination field:

$$
\psi_j = \frac{2\hat{F}_{\mathrm{d}j} - F_{\max} - F_{\min}}{F_{\max} - F_{\min}}. \tag{B2}
$$

In this transformed representation, the values of $\psi$ lie in the range $-1 \leq \psi_j \leq 1$ and the global mean correction is given by

$$
\overline{\Delta \psi} = \frac{2\overline{\Delta F}}{F_{\max} - F_{\min}}. \tag{B3}
$$

A non-dimensional version of $\gamma$ can then be written

$$
\dot{\gamma}_j = (1 - \psi_j^2)^\mu. \tag{B4}
$$

To prevent a corrected field value from exceeding the imposed limits, we require for all $j$:

$$
\frac{\dot{\gamma}_j}{\overline{\dot{\gamma}}}|\overline{\Delta \psi}| \leq 1 - \psi_j \text{ if } \overline{\Delta \mathrm{F}} > 0
$$
$$
\frac{\dot{\gamma}_j}{\overline{\dot{\gamma}}}|\overline{\Delta \psi}| \leq 1 + \psi_j \text{ if } \overline{\Delta \mathrm{F}} < 0. \tag{B5}
$$

Next, excluding all remapped cell values of $F_{\max}$ or $F_{\min}$, we find among the remaining cells the extreme value, $\psi_{\mathrm{e}}$ that, given the sign of $\overline{\Delta F}$, is of relevance:

$$
\text{if } \overline{\Delta \mathrm{F}} > 0, \quad \psi_{\mathrm{e}} = \max[\psi_{\mathrm{j}}] \text{ for all } \psi_j \neq 1
$$
$$
\text{if } \overline{\Delta \mathrm{F}} < 0, \quad \psi_{\mathrm{e}} = \min[\psi_{\mathrm{j}}] \text{ for all } \psi_j \neq -1. \tag{B6}
$$

It can be shown that (B5) is satisfied for all $j$ if

$$\frac{\dot{\gamma}_{\mathrm{e}}}{\overline{\dot{\gamma}}}|\overline{\Delta\psi}| + |\psi_{\mathrm{e}}| \leq 1, \tag{B7}$$

where $\dot{\gamma}_{\mathrm{e}}$ is evaluated with $\psi_j = \psi_{\mathrm{e}}$.

The differences in the corrections made to the collection of cells is minimized when equality holds in (B7), which results in the cell value closest to the extreme being adjusted to equal the extreme value. All other values will be adjusted by a larger amount, but it can be shown that having initially been further from the extreme, no other value will reach or exceed the extreme. It is sufficient, then, to find the value of $\mu$ in (B4) that ensures equality holds in (B7). The problem is nonlinear and $\mu$ must be calculated using an iterative method. An approximate value will be obtained first, followed by iterative application of a formula, which is derived by perturbing the previous approximation of $\mu$.

Substituting the expression for $\dot{\gamma}$ into (B7), we find

$$\frac{\dot{\gamma}_{\mathrm{e}}}{\overline{\dot{\gamma}}} = \frac{(1 - \psi_{\mathrm{e}}^2)^\mu}{(1 - \psi^2)^\mu} = \frac{1 - |\psi_{\mathrm{e}}|}{|\overline{\Delta\psi}|}. \tag{B8}$$

To obtain a first approximation, we note that the global mean quantity in the denominator of the left side of (B8) will not exceed 1 because $-1 \leq \psi_j \leq 1$. If we solve the equation for $\mu$ with the denominator set to 1, we will obtain an underestimate of the true value of $\mu$, but this will serve as a first estimate:

$$\mu_0 = \frac{\ln(\frac{1 - |\psi_{\mathrm{e}}|}{|\overline{\Delta\psi}|})}{\ln(1 - \psi_{\mathrm{e}}^2)}. \tag{B9}$$

The formula used to improve iteratively on this estimate is derived by setting $\mu_{n+1} = \mu_n(1 + \epsilon_n)$ where $\epsilon_n$ is obtained by substituting $\mu_n$ into (B8) followed by a first order expansion for small $\epsilon_n$, which leads to:

$$\epsilon_n = \frac{1 - \alpha_n}{\alpha_n \ln \dot{\gamma}_{\mathrm{e}n} - \frac{\dot{\gamma}_n}{\overline{\dot{\gamma}}_n} \ln \dot{\gamma}_n}, \tag{B10}$$

where

$$\alpha_n = \frac{\dot{\gamma}_{\mathrm{e}n}|\overline{\Delta\psi}|}{\overline{\dot{\gamma}}_n(1 - |\psi_{\mathrm{e}}|)}, \tag{B11}$$

and $\dot{\gamma}_n$ is defined by (B4) with $\mu = \mu_n$. Convergence is reached when, consistent with (B8), $\alpha = 1$, which yields $\epsilon = 0$.

The above method of calculating $\mu$ was applied to the simple temperature example test case in section 3. the values are given in Table 2. For all resolutions considered there, $\mu$ can be determined to three significant figures after two iterations and to five significant figures after three iterations. It can be shown that under this approach with the first approximation calculated using (B9), $\epsilon_n$ will invariably be positive. This means that although each estimate will improve on the previous estimate and approach convergence, $\mu$ will always be an underestimate of the value we seek. This results in a correction to the value nearest the extreme, $\psi_e$, that is slightly too large, which means that the corrected value will exceed the limit imposed on the field. In order to prevent this, one can in (B11) slightly inflate the value of $|\overline{\Delta\psi}|$, thereby forcing the iterative method to converge on a value of $\mu$ that is slightly larger than absolutely necessary. If the inflation factor chosen to multiply $|\overline{\Delta\psi}|$ is not too large (say,

1.02), then the value of $\mu$ will be nearly optimal in distributing the correction as evenly as possible across the domain while ensuring that the limits on the field are respected. In the test example considered in section 3, the difference in corrections with the slightly overestimated $\mu$, compared with the optimal $\mu$, were less than a hundredth of a degree Celsius in the lowest resolution remapping and were smaller at the finer resolutions.

*Author contributions.*  The author was solely responsible for this contribution.

*Competing interests.*  The author declares that he has no competing interests.

*Acknowledgements.*  I thank Paul Durack and Paul Ullrich for their helpful comments on the original draft of this article. I am grateful for the thoughtful comments and suggestions offered by the reviewers and for offline exchanges with M. Hanke and C. Zender who explained certain characteristics of their software libraries and generously shared their considerable knowledge regarding remapping algorithms. This work was performed under the auspices of the U.S. DOE by Lawrence Livermore National Laboratory under contract DEAC52-07NA27344. It was supported by the Regional and Global Modeling Analysis (RGMA) program area under DOE's Biological and Environmental Research (BER) Program.

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

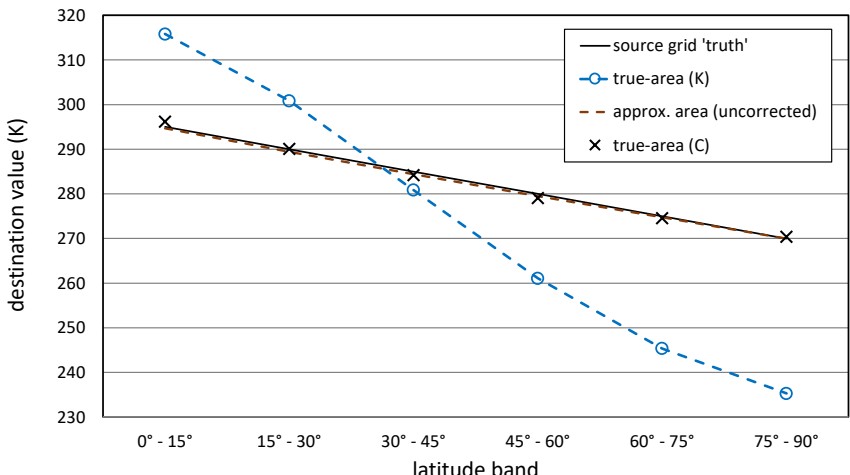

**Figure 1.** For the example described in the text, destination grid cell values resulting from different remapping options. The source data were defined on a longitude by latitude grid of $30°$ by $15°$ resolution and then mapped to a destination grid with half the longitudinal resolution but the same latitude spacing (i.e., $60°$ by $15°$). The solid black line (source grid 'truth') also represents the destination values that would result from correctly remapping the data to the destination grid, applying an algorithm that correctly reconstructed the grid cell shapes.

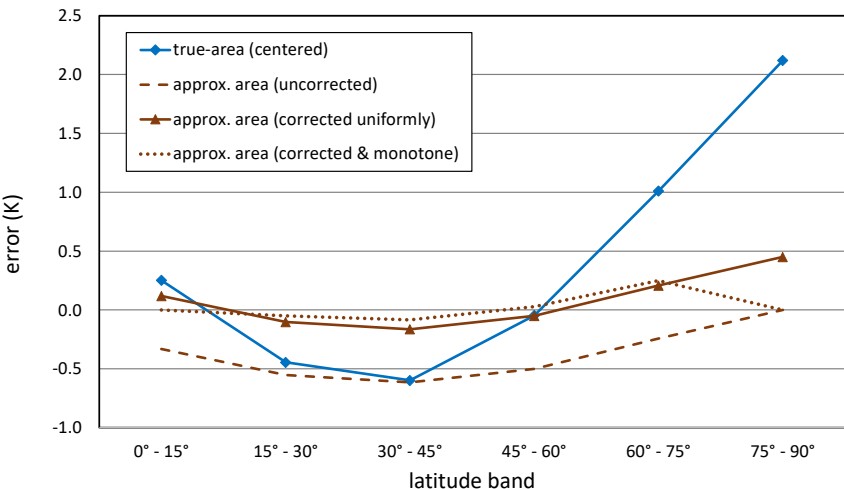

**Figure 2.** For the example described in the text, error in destination grid cell values resulting from different calculational options.

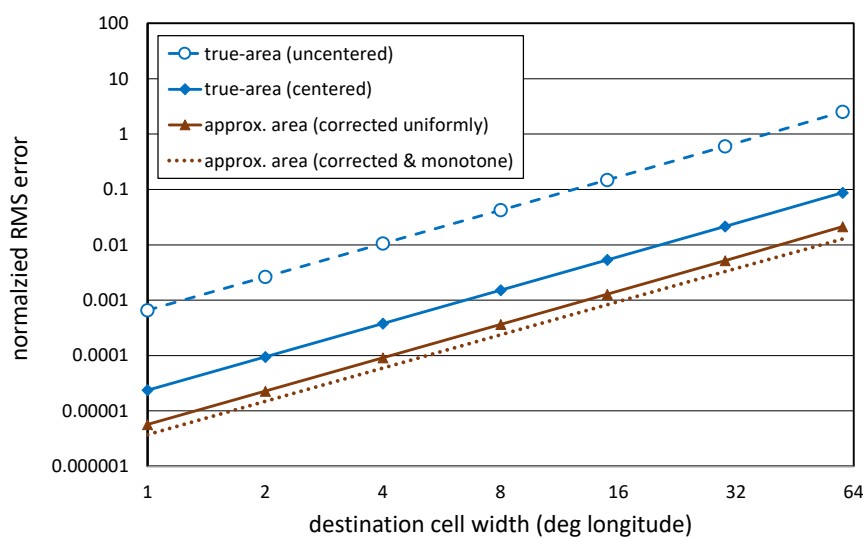

**Figure 3.** For the example described in the text, dependence on the longitude cell width of the RMS error in destination cell values, with the error calculated over all latitudes and weighted by area. Only the remapping options that preserve the true global mean are shown. The RMS errors have been normalized by the true spatial standard deviation of the variable. Expressed in this way, an error equal to 1 means, for example, that the RMS error is as large as the spatial standard deviation of the variable, which in this example is 7.2 K. The mapping is always from a source grid with longitude cell widths half that of the destination grid but the same latitude resolution ($15°$ latitude bands for all longitudinal resolutions).

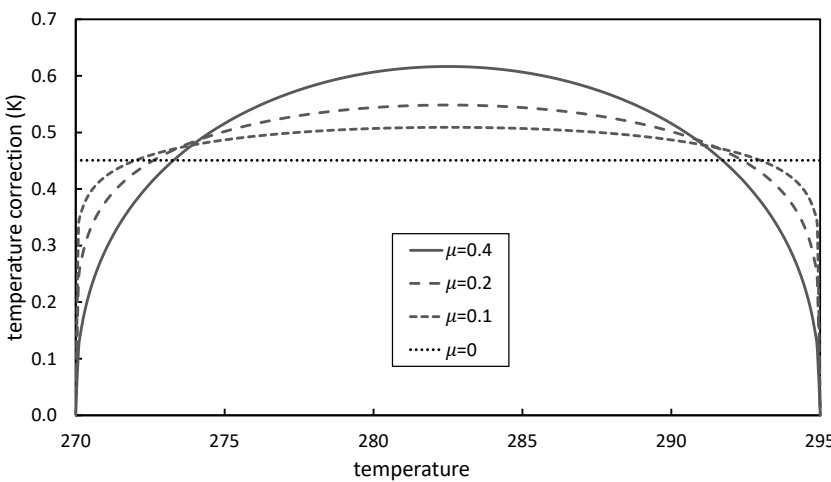

**Figure 4.** For the example described in the text, dependence of the temperature correction, $(\gamma^\mu/\overline{\gamma^\mu})(\overline{\hat{T}_d} - \overline{T_s})$, on temperature and the exponent $\mu$. The same shaped curves apply to any variable with appropriately rescaled axes.

**Table 1.** True cell areas ($A^*$) and approximate cell areas ($A$) for source and destination spherical coordinate grids with longitudinal cell widths of $30°$ and $60°$, respectively. The approximate areas are calculated assuming all bounds coincide with great circles. All areas are expressed as solid angles obtained by dividing the actual cell area by the square of Earth's radius. For ease of comparison with destination cell areas, source cell areas have been doubled.

| | source cells | | destination cells | |
|---|---|---|---|---|
| latitude band | $2A_{\mathrm{s}i}^*$ | $2A_{\mathrm{s}i}$ | $A_{\mathrm{d}j}^*$ | $A_{\mathrm{d}j}$ |
| $0° - 15°$ | 0.271 | 0.277 | 0.271 | 0.297 |
| $15° - 30°$ | 0.253 | 0.256 | 0.253 | 0.265 |
| $30° - 45°$ | 0.217 | 0.216 | 0.217 | 0.213 |
| $45° - 60°$ | 0.166 | 0.163 | 0.166 | 0.152 |
| $60° - 75°$. | 0.105 | 0.101 | 0.105 | 0.090 |
| $75° - 90°$. | 0.036 | 0.034 | 0.036 | 0.030 |

**Table 2.** For the temperature field and various grid resolutions considered here, the $\mu$ values that most evenly distribute corrections needed to preserve the true global mean without shifting destination values outside the source field range. See Appendix 2 for the procedure used to determine $\mu$. The longitudinal resolution of the source field is invariably twice that of the destination field.

| destination cell width | $\mu$ |
|---|---|
| $1°$ | 0.0390 |
| $2°$ | 0.0446 |
| $4°$ | 0.0522 |
| $15°$ | 0.0629 |
| $30°$ | 0.0773 |
| $60°$ | 0.1607 |