# Peer review of "Truly Conserving with Conservative Remapping Methods"

_Geoscientific Model Development, 2023_

## Author Response (AR1)

**Reviewer RC1:**

Karl Taylor has presented us with a very carefully written and clearly explained treatise on the problems we encounter when shifting Earth system modeling data from one grid to another. It is a valuable contribution to Geoscientific Model Development and can be published almost as is. Personally, I would like to see Dr. Taylor address some of the issues that I raise below in his text, not necessarily with new recipes. For example, a very important use for this work (not noted here) is in remapping emissions from some standard regular lat-long grid onto the model grid. (One could argue that radiation is an emission.) It is essential that this remapping conserve emissions, e.g., of greenhouse gases, because otherwise the radiative forcing and climate trajectories will diverge based on model grid. This is a long-standing problem that many have solved by some but still plagues CESM versions and its related E3SM codes.

I have added a couple of sentences regarding the importance of conserving trace species and associated sources and sinks to the introduction. I also now mention this in the second paragraph of the conclusions.

L3: Excellent and agreed; yet, there is a more difficult problem of the convergence/divergences that must be preserved when winds or fluxes are remapped. Do you have a solution for that?

No, but I think conservative remapping by itself is clearly not the answer. That problem is outside the scope of the current article.

L31: The assumption that there is a single $r_s$ or $r_d$ is dangerous. The Earth is not a sphere and more accurate models may need to take that into account. Moreover, surface orography is standard in most models -- how is that treated in this equation? If you are regridding a quantity at the top of the atmosphere, a geometric atmosphere (spherical expansion, g varies with altitude) should probably be accounted for.

Is this problem accounted for somehow in discussion re equation 8?

I think models will for many years to come continue to make the spherical earth assumption because the errors due to this approximation must be dwarfed by other model errors. Also, the the equations governing flow on an oblate spheroid are much more complicated. (See Gates, W. L., 2004: Derivation of the Equations of Atmospheric Motion in Oblate Spheroidal Coordinates. J. Atmos. Sci., 61, 2478–2487, https://doi.org/10.1175/1520-0469(2004)061<2478:DOTEOA>2.0.CO;2. For certain observations, the earth is not assumed to be a spheroid, but again I think the use of such observations (given their uncertainty) in conjunction with models (given their uncertainties) doesn't warrant taking this into account.

As to the expansion of the sphere in going from the surface to the top of the atmosphere, conservative remapping of a field at any altitude only depends on the ratio $r_s/r_d$, and this does not depend on altitude.

L141ff:  The analysis now gets into some very interesting problems.  It is nicely explained how rescaling to get global conservation gets one into trouble in so many ways.  That this problem is still with is indeed disconcerting.  When we were developing tracer transport algorithms many decades ago, conservation was an important but most often violated condition.  The approach of many was simply to evaluate the numerical loss and add it back into the system – but of course, there was no information on where to put it, and uniform replacement created unphysical variability in the chemical tracer.  This seems to be going on here.  The other test we used, similar to the one here was to transport a uniform tracer and show that artificial variations did not occur.

Yes indeed, these problems have been around for decades.  Many modelers understand the issues and have devised workarounds.  When a regridding package can accurately represent cell shapes, the problems largely disappear, so we should encourage the generalization of these packages to handle all common grids accurately.  Most analysts, I think, are not aware of these problems, and I hope this article will be helpful in that regard.

L205: This case and the problems with the remapping are clearly stated, thanks.  Do you have other examples of problems with cubed-sphere or polygonal grids that are being used so much in high-resolution models?

I now note at the end of the first paragraph of the conclusions that cubed-sphere grids will not be mapped accurately if a regridding package assumes cell edges coincide with straight lines on a equirectangular projection (as in SCRIP).

L265ff:  Indeed there are many cases re emissions and chemistry where this would be a show stopper. I am not sure that the chemistry community would be happy with these mu factors (eqn 19):  they seem arbitrary and would have to be tuned for every species and variable (emissions, concentrations, ….).  Would we just be adding many other parameters to be tuned?

If it is important to preserve the true global mean, some sort of adjustment must be made, which if not overly computationally expensive and may need to be somewhat arbitrary and ad hoc.  One might, for example, simply apply a uniform adjustment and then clip any value deemed unphysical (or out of bounds).  The clipping would, of course, prevent the correction from perfectly conserving, so one would have to iterate until no clipping were needed.  (To ensure convergence, slight "over-clipping" would be required.). This process would have to be applied for every new field treated (and obviously for every species).  I have suggested an alternative that distributes the correction according to the function given by equation 19.  In place of the iteration required in uniform adjustment followed by clipping, my approach requires iteration to determine the value of mu that distributes the correction as evenly as possible across the field, while preventing values from exceeding imposed (or physical) limits. In the test case considered, only a few iterations are needed, so the method is likely to be computational no more expensive than other options.

L306ff:  OK, the recipes are clear.  Do you have any recommendations if this had to be applied 100+ items at a time?

I'd like to not get into this.  The recommendations would be highly dependent on the case, and I think it should not be too difficult to suggest a specific recommendation without discussing a number of different possible code architectures.

L370:  Yes, indeed, there is much to do with column integrals being remapped (e.g., ozone hole) in additions to the mass in each layer.  Any example under (g) below?  OK, I see this comes in Section 5 later.

Yes.

L453ff:  What if we finally go to geometric grids where g is the true value, function of R+z, and is not constant.  Also, how do you average horizontally using cells with different lower boundaries?  Is that obvious from above?

I think addressing the variation of g with altitude or location will require a number of changes to models and analysis procedures but is currently not of much interest.  I would rather not speculate about how that might impact remapping procedures.

As for horizontal averaging over cells of different thickness as it relates to vertical interpolation, the problem isn't limited to the lowest layer.  In the "recipe" section 4, the "possibly troublesome 3-d variables" discussed in items e, f and g all involve horizontal remapping of a field of variable thickness.  I hope this provides understandable guidance on how to handle this problem.

L487: This is great, but if you moved emissions from equator to pole, you can truly make a mess of the results.  Can you invoke regionally conservative mapping, not just global?

I have not found a way to do this except to ask those creating the weights to generalize their packages so the user can specify how the edges of grid cells are defined and then use that information to correctly derive the weights.

**Reviewer RC2 (Vijay Mahadevan)**

The paper is well written and coherently explains about conservation principles in remapping climate fields between meshes on the sphere, with potential pitfalls in existing remapping libraries and ways to address them. This are some valuable discussions with simple equations and recipes. I do have some comments listed below and I think it would help the clarity of the manuscript if they can be addressed well.

1. Line 75: The usual assumption is that the cell edges run along great circles even though for a cartesian longitude by latitude grid, the latitude cell bounds follow latitude circles, not great circles.

   Does this mean that just representation of RLL meshes are represented incorrectly? Or is this a general statement for arbitrary unstructured mesh representations on the sphere.

   I have rewritten the paragraph to clarify that existing algorithms make assumptions which when the algorithm's assumption is inconsistent with the grid construction can lead to errors.

2. In the paragraph starting at Line 189 that discusses results shown in Figure 2, I do not see a description of the case with $\mu=1/4$. Please explain what the "approx area (corrected $\mu=1/4$)" refers to. This is not introduced and discussed until Line 270. Please fix this inconsistency as it will confuse the readers.

   I have added a sentence to this paragraph to inform the reader that this case (the dotted line) will be discussed in section 3.

3. Line 280: As a rule in calculating $\gamma$, a relatively small value of $\mu$, say 0.1 or 0.25, should be tried first, and then the destination values should be checked to confirm that none exceeds the extremes found in the source field.

   If $\mu$ is a heuristic that is determined by trial-and-error, how can this be more practically used in a realistic remapper? One way is to make it into a nonlinear bound-constrained minimization procedure to find $\mu$ such that field bounds are preserved with a minimal value of $\gamma$. That should yield an optimal value of $\mu$ for the remapping procedure. However, this implies a global minimization procedure, which in general can be expensive for high-res cases.

   Following your suggestion, I have now included Appendix B where I detail how the minimum $\mu$ can be determined that distributes the correction as evenly as possible across the field while constraining all values to remain within the maximum and minimum specified. The iterative scheme to determine mu converges after only a few

iterations and each iteration only requires two sums over all destination grid cells which should not be too computationally burdensome.

4.  Line 307: Some conservative remapping packages (see Appendix A) have not been designed to handle the most general cases considered here (e.g., fields with missing values or grid cells that are partially masked).

    I believe the production ready remapping packages like ESMF, TempestRemap and SCRIP (OASIS3-MCT version). It would be helpful to verify this statement, and cite documentation that provides details on whether these features are unsupported for the packages.

    It is possible that some of the major packages in fact support this, but I have found it impossible to understand the documentation of those packages to the extent that I would trust them to do it correctly.   Especially when the mask is not a binary mask, the packages may require the user to apply a pre-processing step (multiply field values by an area fraction) and a post-processing step ("undoing" the fractions).   Given the insufficiently clear documentation (I think with the current available documentation, many climate researchers analyzing model output using some of the packages would have a difficult time correctly conserving when cells are partially masked), I think I should stick with my statement with the slight modification of replacing "have not been designed" with "may not be designed or may not clearly document".

5.  Equation 21 describes the "consistency" property, which guarantees that a constant field remapped from source to target remains the same, and that the projected field in general converges at the theoretical rate asymptotically under uniform source and target mesh refinements. It would be good to state that explicitly.

    I have now included a sentence under the equation (now equation 22) stating this: "If (22) is satisfied, the remapping algorithm will be "consistent" in the sense that a spatially constant source field will remain spatially uniform on the destination grid."

6.  Paragraph starting at Line 360 discusses a sub-cell approach to compute a more accurate remapping method. You are essentially suggesting the use of a uniformly refined source/destination mesh in the computation and then aggregate the field data onto an embedded coarser mesh. But in addition to increasing the actual cost of computing the mesh intersections between source and destination meshes for remapping, I do not see how this will improve the accuracy of the remapped field itself if the source field is defined on the coarse source mesh. If the premise is that smaller cells yield smaller area-errors in calculations, one could always compute the intersection cells using the actual coarse source/destination grids, but use a sub-cell approach on the intersection (overlap) mesh to get a better area estimate, which directly affects the weights. In either case, I do not feel there is substantial value in this discussion without

any particular proof to support the conclusion or an experimental confirmation of the behavior.

I have come around to your view on this and have deleted this paragraph.

7. The lines starting Line 370 should probably be identified as a subsection with an appropriate title. A broad section title of "Recipes for regridding" may not be fully appopriate to the discussion from Line 370-441. Additionally, while you make relevant points, it reads like a bucket of observations that sway from vertical remap to preserving various forms of ratios with many similar underlying ideas. I suggest the author rephrase this section and describe the details more concisely as it reads more verbose than it needs to be.

My intention in this section was to provide a clear and detailed description of how best to conservatively remap a variety of different variables on 2-d grids.  The first part describes the "standard procedure" in detail and the second part how it gets applied/modified for specific variables.  I think analysts will appreciate having this detailed guide, so I'd like to keep it as a single cohesive section.  In the itemized list, I have now set the names of specific variables in italics so a reader might easily skip any variable of no interest.  (Note that vertical remap is not covered here but in a subsequent section.)

8. Line 445: If a field will ultimately be mapped to multiple horizontal destination grids, then performing the vertical interpolation first will usually reduce the computational expense.

This really depends on the number of vertical layers in the source and destination grid. I suggest rephrasing this statement as a condition based on vertical resolution. In this context, I am curious whether a 2Dx1D vs a 1Dx2D tensor product would yield better field accuracy and conservation. Or do you think this should not be a factor as the error components in (x,y) and z are decoupled here? Furthermore, it will be valuable to make a statement whether a tensor-product approach will suffice for practical 3D coupling problems or if full 3D remapping algorithms will be necessary. For extruded data, which you have assumed in this section, it might make sense to use 2Dx1D approaches, but for more general settings, this may not yield the right results.

For a single 3-d conservative remapping, yes, it will depend on the vertical and horizontal resolutions and by how much the resolution is coarsened in each direction when remapping.  I have elected not to get into the trade-offs because it is tangential to the discussion.  I no longer claim that interpolating in the vertical first will reduce the computational burden.

I have now included a sentence suggesting that the sequential application of independent vertical and horizontal remappings may not always be sufficiently

sophisticated when coupling component models. Discussing the various options for doing a fully 3D mapping, however, is beyond the scope of this article. In any case, I do not feel qualified to make a statement along the lines suggested; I would have to do considerable research to comment intelligently.

9. It would add value to the manuscript to discuss a section about potential extrapolation of data in both 2D and 3D remapping scenarios, and potential methods or fixers to recover conservation (if at all possible). In a way, this is also very relevant to your comments on the current limitation of remapping libraries in dealing with partially unmasked cells.

   These are interesting questions, but I think they exceed the scope of the present article, which focuses on the application of currently available weight-generators for 2D grids. I also do not have enough expertise about such methods to intelligently discuss the issues.

10. Line 489-491: Consider, for example, the geopotential height and wind fields carried on a relatively coarse source grid. If these fields were mapped conservatively to a much finer resolution grid, box-fill contour plots of the resultant fields would look like slightly blurred versions of the box-fill plots of the original fields.

    This is a common issue when using conservative coarse-to-fine projections, and is often referred to as the mesh-imprinting phenomena.

    Thanks for bringing this to my attention. I have added, parenthetically, "(often referred to as the mesh-imprinting phenomena)".

11. In terms of the "Appendix A: Remapping packages", I would like to suggest one of the remapping packages that has been developed recently at Argonne National Laboratory [1]. While it uses TempestRemap for weight computation, the mesh description and intersections are computed natively in the MOAB library. Disclaimer: I am the lead developer of this package and while I usually do not recommend my papers in reviews, it seems quite relevant here.

    [1] https://doi.org/10.5194/gmd-13-2355-2020

    Thanks for this reference. I now include in Appendix A the MOAB library.

12. I also want to note that the fixers used to recover the conservation and monotonicity are similar to the Clip-And-Assured-Sum (CAAS) [2] approach that were introduced to recover property-preservation and applied successfully to several experiments [3] and some ongoing production work as well. It may be very valuable to compare and contrast how both these approaches recover and/or differ in terms of accuracy, while preserving monotonicity and conservation without recomputing weights since the procedures for

recovery inherently introduce a nonlinear element when dealing with linear maps.

[2] https://doi.org/10.1137/18M1165414
[3] https://doi.org/10.5194/gmd-15-6601-2022

I have mentioned now that there are other approaches one might follow in making the correction and referenced a small selection of these.   I, however, think that including a comparison like the one suggested would be difficult to generalize to any kind of conclusion because the result must surely depend on what the source field looks like.  I therefore think that it is sufficient here to make the point that there are global corrections that need to be made and that they should preserve certain properties of the original field.

13. There are several other minor technical revisions that will help the readability of the paper as well. I have not listed them all here.

I have made many minor revisions to the text that I hope will make it more readable.

**Reviewer RC3:**

This is a well written paper that clearly explains some technical issues related to conservative remapping of climate data. It is of most relevance to remapping data from unstructured model grids to lat/lon grids commonly used by analysis packages. It highlights a recent issue driven by the fact that nearly all modern global models have transitioned to unstructured grids with cell boundaries give by great circle arcs, while lat/lon grids have cell boundaries given by lines of constant latitude or longitude.

Conservative maps for these applications are almost always based on the incremental remap approach (Dukowitz & Kodis, SIAM J. Sci Stat.Comput. 1987) which requires computing intersections of the cell boundaries. Most remap packages compute these intersections assuming both grids use great circle arcs, or both grids use lines of constant lat/lon (in the case of SCRIP)

General comments:

Section 2: The area correction approach described here, as well as its many issues, is well known to coupled model developers, since they need to remap fluxes between different model components. MCT and ESMF based couplers have been applying one form of this correction since the introduction of model components using great-circle-arc bounded cells. But the technique and its implications (and in particular, violation of bounds preservation) may not be widely known in the analysis community and thus I think the presentation here is useful. I'm assuming that this corrections described here have been published before, but I dont actually have a good peer reviewed paper to suggest. However, for example it is described in "Area Correction of Fluxes" in the CESM CPL7 user's guide.

I too have been unable to find a clear, published descriptions of exactly what different regridding packages do. That was one motivation for writing this paper. The description in the CESM CPL7 user's guide indicates that a correction is made (and what input is needed to make the correction), but not exactly how the correction is calculated. Other documentation I think is similarly inadequate for enabling the average reader to write an algorithm to reproduce what the "black box" regridding package does.

Section 3: The author describes an ad-hoc procedure to improve bounds preservation properties while conserving global means, through the use of a correction coefficient gamma an associated parameter mu. There are better mathematical solutions to this problem, which are guaranteed to conserve means and bounds such as CAAR and QLT. QLT has the further advantage that is minimizes the amount of non-locality of the adjustments. Are there any advantages to the procedure suggested here? If so they should be mentioned. (echoing the comments of one of the other reviewers, who provided the relevant references: https://doi.org/10.1137/18M1165414, https://doi.org/10.5194/gmd-15-6601-2022). GMD

I now cite examples of more sophisticated filtering methods that ensure monotonicity, but my intent in proposing an alternative is not to propose a better correction but, rather, a simpler correction, which the typical climate data analyst could easily code up.  For purposes such as coupling model components, one of several alternatives might be more accurate, but, perhaps, more computationally expensive.

Minor revisions:

1. The paper implies existing remapping approaches use great circle arcs, but SCRIP can use straight lines in both lat/lon and Mercator projections, so this should be mentioned.

If I correctly interpret the SCRIP documentation available at https://github.com/SCRIP-Project/SCRIP/tree/master/SCRIP/doc, the cell edges are defined by equations of the form theta-theta1 = alpha*(lambda-lambda1), where alpha = (theta2-theta1)/(lambda2-lambda1).  It follows that these would appear as straight lines on an equirectangular projection, rather than on a Mercator projection.  That being said, a latxlon grid would be a special case with "rectangular" grid cells with two sides coinciding with lines of constant latitude and two side coinciding with lines of constant longitude.  I now indicate in section 2 (in the paragraph after equation 6) that some packages assume cells follow great circles and others assume they appear as straight lines on an equirectangular projection.  I also point this out in the Appendix when describing the SCRIP package.

2. Section 2: It would be worthwhile to search for previous work describing this area ratio procedure and give those authors credit.

 If by "area ratio procedure", you mean equation 4, I have failed to find in published works this specific form.  All first-order conservative schemes, I think, rely on this method, but in their descriptions, I do not find an appropriate reference.  If I am missing a reference that should be included here, I would be happy to learn of it.  Elsewhere I now reference Dukowitz and Kodis (1987), who originally proposed how to compute conservative weights.

3. Add descriptions to the CAAR and QLT approaches and compare with the gamma/mu approach presented here.

The intent here is not to suggest a superior method for making a correction (needed to preserve the global mean or integral) but, rather, to enable the typical climate data analyst to be able to conserve when remapping data.  For such purposes, there is little need to do anything too sophisticated, so I think it would be of little value to discuss such approaches as suggested above.

**Reviewer CC1 (Moritz Hanke):**

Hello,

You hopefully do not mind, if I give some comments and remarks on your paper.

Your comments have been valuable, and I think they have measurably improved the paper. Thank you for them.

Thank you very much for this paper! It truly helps with implementing coupling software for climate models and gives great guidelines for users on how to use conservative remapping.

My experience with conservative remapping is limited to ESMF, SCRIP, and XIOS, whose implementations I read at some point. In addition, I implemented the conservative remapping algorithm in YAC.

Regarding the misrepresentation of true grid cell shapes:

- To my knowledge it is true, that ESMF represents all edges using great circles, which leads to the issues described in this paper.
- In SCRIP, vertices are connected using rhumb lines, which results in the correct representation of edges located on circles of longitude and latitude, but is wrong for edges on great circles. Due to the way intersections and areas are computed in SCRIP, accuracy can become very low towards the pole. Therefore, the library has the option to project cells, which are close to the pole, towards the equator. This in turn changes the representation of the edges to something else. It also produces a lot of other issues (see [1] and [2]).

  Thanks for introducing me to "rhumb lines", which I was not familiar with. I think that SCRIP, however, assumes cell sides would appear as straight lines on an equirectangular projection (aka, equidistant cylindrical projection), which do not coincide with rhumb lines (which appear as straight lines on a Mercator projection). I hope I am not misinterpreting the SCRIP documentation. In either case, of course, a cell side connecting two vertices as the same latitude would fall along a latitude circle (and similarly for longitude). I have now noted in section 2 that some packages assume great- circles and others straight lines on equirectangular projection, and in the Appendix I describe the method used in SCRIP.

- As for ESMF, XIOS also only uses great circles to represent edges. However, it approximates edges on circles of latitudes (and small circles, if I remember correctly) using small great circle segments, which should significantly reduce the error in the misrepresentation of the grid cell shapes.

Thanks for bringing this to my attention. I now point out in the Appendix this "trick" that XIOS relies on to improve its accuracy.

- YAC supports edges on great circles and circles of longitude/latitude. Therefore, it should not misrepresent the shapes of grid cells, unless the edges are on different circles than ones already mentioned.

Thanks for providing this definitive information, which I have included in the description of YAC in the appendix and also noted in the text, since YAC seems to be the only remapping package with this capability (see the new draft, line 85).

Since, half of the implementations for conservative interpolation, which I personally know, more or less correctly represent the the true grid cell shapes for most common grid types, I would not agree with the statement: "many (perhaps all) of these packages slightly misrepresent the true grid cell shapes for certain common grids".

I've revised this to read: "It should be noted that nearly all of these packages slightly misrepresent the true shape of grid cells found in some subset of commonly encountered grids"

As best I can determine, only YAC can accurately generate weights when one grid is a regular latxlon grid and the other comprises cells with edges following great circles. (I now understand that the XIOS code includes a clever method for reducing the size of the error for regular latxlon grids, but there will still be some residual error.)

Line 75-78:

This is true for ESMF. SCRIP will assume the correct shape for regular lon/lat grids, if no tranformation for cell close to the poles is applied. XIOS and YAC will also assume the correct shape, if the grid is defined as a regular lon/lat grid (also works for Gaussian reduced grids).

Based on this information, I have now specifically noted the capabilities and limitations of the various packages in Appendix A.

Line 110-119:

Great quantification of error in the misrepresent of the grid shape.

Line 150:

It could be noted that the computation of the global mean, may be time consuming and have bad scaling behavior. However, since regular lon/lat grids are in my experience seldomly used for high resolutions runs, this might not be an issue.

This is true, and the proposed adjustment is subsequently not recommended. Later, however, there are similar sums required in applying other methods for correcting the values so that the global mean/integral can be preserved.

Line 243:

Recent releases of YAC contain a feature, which allows the user the provide fractions along with the source field for the interpolation (see [3]). These fractions are taken into account as described in formula (16) of the paper. My implementation was based on the work of the OASIS development team. A similar feature will most probably be part of the next OASIS release.

That is good news. In the paragraph following equation (16) I now indicate that some existing packages can remap fields with cells fractionally masked.   My aim here was to clearly describe the method whereby weights should be applied so that users could code up the procedure themselves if they chose to.

Line 354 "2) the unmasked fractions on the destination grid are defined by (15)":

This sums up an important point for users that set up the grids and masks for two coupled models!

Yes indeed.

Paragraph starting at line 360:

If the remapping algorithm supports concave cells, an alternative approach would be to add additional vertices along the edges located on latitude circles (as done automatically by XIOS). This should produce the same results as using sub-cells, but directly produces weights that can be used without the need for any post-processing of the weights. A drawback of this approach may be, that the remapping algorithm might support concave cells by triangulation, which could reduce accuracy.

This is worth exploring, but I have been unable to find a way to concisely describe the idea without disrupting the flow of text, so I hope that it is acceptable to leave this to others.

Paragraph starting at line 471:

Alternatively, if the source fractions are changed, the source model can multiply the source field with the source fractions and interpolate the source fractions along with the source fields. The target model can then adjust the received field using the received fractions.

Here I am referring to the straight-forward application of most remapping packages.  I agree that their weights (not accounting for fractional masking) can be used with the masked fractions to remap data conservatively.  That is what is achieved by following the recipe I

propose.  For some packages (and YAC may be one of these), the package may include options that already facilitate application of the recipe proposed here, but I was unable to determine definitively if this is the case.

Line 495-497:

What about second order conservative remapping (implemented in ESMF, XIOS, and YAC based on [4]).

Yes, that should better preserve the approximate balances. I now include a sentence at the end of the paragraph pointing this out.

List entry starting at line 512:

In CDO, only the implementation of conservative remapping is based on YAC.

Thanks for this information.  I have now noted this.

List entry starting at line 518:

MCT itself does not have any weight computation functionality, therefore it probably does not fit in this list.

If that is the case, I agree, so I have removed MCT from the packages listed.

List entry starting at line 524:

OASIS either gets its weights from offline weight computation tools (e.g. ESMF) or it can generate weights for conservative interpolation using SCRIP. MCT is used for the parallel sparse matrix vector multiplication, but not for the weight computation itself.

I have revised the text to be consistent with this.

List entry starting at line 528:

ESMF, XIOS, and YAC use a significantly different algorithm from SCRIP. Therefore, I would not agree that "most remapping packages have been based on SCRIP".

I have deleted this assertion.

General remarks:

- CERFACS did a benchmark for regridding libraries, which might be interesting (see [5]).

Thanks for reminding me of this study. I now cite it this.

- Regridding libraries usually provide two normalization options for the first order conservative remapping (destarea and fracarea). If I understood the paper correctly, all formulas assume that destarea is being used for the weight computation. You could also discuss the implications of the fracarea normalization method.

  The approach suggested here is to obtain the weights generated by remapping packages for the case of no missing/masked variables. I think in this case the weights returned are identical (independent of the destarea/fracarea directive) and can then be applied to fields where cells are individually wholly or partially masked. Please let me know if I am wrong about this. Part of what motivated me to look into this was that I found the treatment of masked fields by most packages very confusing, and for time-dependent masks it seemed like the simplest application of remapping packages required multiple recalculations of the weights (as the mask evolved).

- Is it possible to also apply the findings of this paper to the second order conservative remapping algorithm described in [4]?

  I think that the procedures recommended in section 4 could be applied to any interpolation procedure, but depending on the scheme, the errors in conserving the true global mean (and integral) might be quite large, which would place a heavy burden on the "correction" step, which might then degrade the accuracy significantly. I am reluctant to speculate about this without further study.

Thanks again for the great paper!

You are most welcome. Thank you for your corrections of several errors.

With best regards,

Moritz Hanke

[1]: Valcke, S.; Piacentini, A. Analysis of SCRIP Conservative Remapping in OASIS3-MCT—Part A, Technical Report TR/CMGC/19-129, CERFACS, France. 2019. Available online: https://oasis.cerfacs.fr/wp-content/uploads/sites/114/2021/08/GLOBC_TR_Valcke-SCRIP_CONSERV_TRNORM_partA_2019.pdf (accessed on 18 January 2022).

[2]: Jonville, G.; Valcke, S. Analysis of SCRIP Conservative Remapping in OASIS3-MCT—Part B, Technical Report TR/CMGC/19-155, CERFACS, France. 2019. Available online: https://oasis.cerfacs.fr/wp-content/uploads/sites/114/2021/08/GLOBC_TR_Jonville-SCRIP_CONSERV_TRNORM_partB_2019.pdf (accessed on 18 January 2022).

[3]: https://dkrz-sw.gitlab-pages.dkrz.de/yac/d0/daa/frac_mask_desc.html

[4]: Kritsikis, E., Aechtner, M., Meurdesoif, Y., and Dubos, T.: Conservative interpolation between general spherical meshes, Geosci. Model Dev., 10, 425–431, https://doi.org/10.5194/gmd-10-425-2017, 2017.

[5]Valcke S, Piacentini A, Jonville G. Benchmarking Regridding Libraries Used in Earth System Modelling. Mathematical and Computational Applications. 2022; 27(2):31. https://doi.org/10.3390/mca27020031

**Reviewer CC2 (Charles Zender):**

I see that many others have already given valuable comments on your manuscript.
Here are my (hopefully non-duplicative) thoughts on your manuscript:

1.  It's a clear, and long-overdue, explication of some of the sausage-making that goes on with weight generators. Good job identifying the need for a clear-eyed analysis of the effect of polygon edge type and methods to minimize associated biases.

2.  My main substantive comment is that the manuscript does not sufficient exhibit the behavior of biases as grid resolution increases. 15 degrees is OK as a starting point, but few people use 15-degree resolution data anymore. Readers would be well-served by high-resolution (e.g., 1x1 or 0.125x0.125 degree) counterparts (perhaps in an appendix) for some of the key figures. Otherwise the casual reader could be left with the mistaken impression that their regridded data is untrustworthy. Providing, in the text, the algebraic factor by which errors scale is no substitute for a few good figures :)

Good point.  Now, before considering the admittedly unusually coarse resolution illustrative example focused on in the manuscript, I point out in the paragraph (beginning at what is now line 120) that the problems and errors associated with remapping that are so evident at coarse resolution are much smaller at fine resolution.  This will hopefully make it clear and reassure readers that the errors may be small enough to be inconsequential in some studies.

I have also noted that errors generally decrease with the square of the grid longitude resolution (second to last paragraph of section 2).  I have pointed out that the case considered (destination grid longitude spacing doubling from 30 deg to 60 deg) is coarser than usually encountered, and for cells remapped from 2 deg to 4 deg, the errors are reduced from about a half degree K to a couple of thousandths of a degree.

Finally, in a concluding paragraph I now remind the reader that the size of the errors featured in the paper would be much smaller when dealing with grids of finer resolution.  I hope that this will again prevent casual readers from concluding that their (almost) conservatively regridded data should be dismissed; I hope they instead will understand that only if they are relying on very coarse grids or require very accurate conservation do they need to be concerned.

3.  p. 4 line 104: The use of "approximate" here seems overly broad. We usually consider the "approximate" shape to be equal to the true shape for all grids except rectangular lat-lon grids.

I have revised the sentence to end with the phrase: "which are based on cell shapes constructed by the remapping algorithm that in some cases are only approximate (e.g., when cells are assumed to be circumscribed by great circles, but in fact have edges following lines of longitude and latitude)."